# Systematic and Narrative Review of the Mediating Role of Personal Relationships Between Mental Health and Nutrition

**DOI:** 10.3390/nu17142318

**Published:** 2025-07-14

**Authors:** Aránzazu de Miguel, José Ángel Agejas, José Víctor Orón

**Affiliations:** 1Department of Gastronomy, School of Law, Business and Government, Universidad Francisco de Vitoria, 28223 Pozuelo de Alarcón, Spain; arantxa.demiguel@ufv.es; 2Department of Ethics, School of Humanities, Universidad Francisco de Vitoria, 28223 Pozuelo de Alarcón, Spain; j.agejas@ufv.es; 3Faculty of Health Sciences, Universidad Francisco de Vitoria, 28223 Pozuelo de Alarcón, Spain

**Keywords:** interpersonal relationships, diet, mental health, mediating effect, well-being, nutrition

## Abstract

Background/Objectives: The complex relationship between nutrition and mental well-being has been widely documented, with empirical evidence supporting both the influence of dietary habits on psychological health and, conversely, the impact of mental state on food choices. However, a critical gap remains in understanding the mechanisms underlying this interaction. While previous literature have examined various biological and psychological mediators, limited attention has been given to the potential mediation role of interpersonal relationships in shaping this dynamic. This article seeks to address this theoretical gap by exploring how the quality of social interactions—both in their intentional dimension and in their lived experiential aspect—may function as a key mediator between nutrition and mental well-being. Methods: Methodologically, a combination of systematic review (PRISMA) and narrative review was employed, given that the findings of the systematic review alone are insufficient to formulate a hypothesis that requires interdisciplinary dialog in a still emerging and underexplored field. Results: The hypothesis we aim to propose is whether the quality of interpersonal relationships acts as a catalyst and enhancer of the effect of nutrients on mental health. Conclusions: This could open new avenues for the design of dietary interventions and mental well-being programs from a socially integrated approach.

## 1. Introduction

Our study is framed within the field of public health, in a context where in recent years, national and international health promotion agencies have expressed growing concern about the sustained increase in obesity rates, largely attributable to the adoption of unhealthy dietary patterns [1]. Simultaneously, there has been a significant rise in the prevalence of mental health disorders, including anxiety, depression, and stress [2]. Both phenomena have a direct impact not only on the financial resources allocated to healthcare systems but also on the urgent need to develop effective interventions that promote holistic well-being.

This dual scenario has generated increasing interest within the scientific community, which in recent years has focused efforts on investigating the potential interrelationships between these two phenomena, as will be illustrated in the studies analyzed below. Several studies have demonstrated a link between diet quality and mental health, contributing to the consolidation of an interdisciplinary field at the intersection of nutrition, psychology, and public health [3,4,5,6,7,8,9,10,11].

An increasing body of high-quality evidence supports the hypothesis that diet quality plays a pivotal role in mental health, contributing to the emergence of nutritional psychiatry as an interdisciplinary field bridging nutrition, psychiatry, and public health. Moving beyond general associations, recent meta-analyses and systematic reviews have delineated the effects of specific dietary patterns—particularly those rich in anti-inflammatory and nutrient-dense components—on psychological outcomes such as depression, anxiety, and cognitive function [3,4,5,6,7].

For instance, one meta-analysis of randomized controlled trials demonstrated that dietary improvement interventions significantly reduced depressive symptoms, yielding moderate effect sizes, especially among individuals with clinically diagnosed depression. These interventions—often based on Mediterranean, anti-inflammatory, or nutrient-rich dietary frameworks—showed efficacy regardless of weight change, suggesting that their beneficial effects are likely mediated through mechanisms independent of body mass regulation [1]. Similarly, a large observational study found that high adherence to healthy dietary indices, such as the Mediterranean or DASH diets, was significantly associated with a lower risk of developing depressive symptoms over time [4]. These findings reinforce the potential of diet quality as a modifiable determinant in the prevention of mood disorders.

Further supporting this association, another study reported that adherence to “healthy” dietary patterns—characterized by a high intake of fruits, vegetables, whole grains, and fish—was inversely related to depression risk. In contrast, “unhealthy” dietary patterns rich in red meat, refined carbohydrates, and sugars were positively associated with depressive outcomes [5]. The proposed mechanisms for these associations include reduced systemic inflammation, attenuation of oxidative stress, regulation of gut microbiota composition, and enhancement of monoaminergic neurotransmission [6].

The relationship between diet quality and mental health also appears to be dose-dependent. A dose–response meta-analysis revealed that incremental improvements in dietary quality were proportionally linked to reduced depression risk, underscoring the importance of sustained dietary adherence over time rather than episodic or short-term changes [7].

Additionally, the mental health benefits associated with dietary patterns such as the Mediterranean diet have been consistently replicated across various populations and age groups. For example, two studies have demonstrated that adherence to Mediterranean dietary principles correlates with both reduced prevalence and incidence of depressive symptoms in adult women [10,11]. Complementary findings in pediatric cohorts suggest that early-life adherence to healthy dietary patterns may foster emotional and cognitive development, with implications for long-term mental health trajectories [8].

One observational synthesis further reinforced the ecological validity of these associations by documenting consistent links between healthy dietary patterns and lower depression risk across diverse cohorts [9]. Collectively, this body of literature substantiates the salutary role of specific dietary frameworks—particularly Mediterranean, anti-inflammatory, and nutrient-dense diets—in supporting mental health. These findings advocate for the inclusion of nutritional strategies in psychiatric contexts and public health agendas, positioning diet as a foundational component of both preventive and therapeutic models of care.

Moreover, a recent review highlights the therapeutic significance of diet as a modifiable factor in the management and prevention of mental disorders. The authors synthesize evidence showing that antioxidant-rich and anti-inflammatory dietary patterns—most notably the Mediterranean diet—may influence neurobiological systems involved in mood regulation and cognitive function. They argue for the incorporation of nutrition into clinical mental health frameworks and public health policy, elevating diet to a clinically relevant element of mental healthcare interventions [12].

Finally, one of the most recent contributions to this field identified a statistically significant inverse association between moderate consumption of polyphenol-rich beverages and symptoms of depression and perceived stress. Conducted in a Mediterranean population and using robust multivariate analyses, the study found that beverages such as espresso coffee, tea, and red wine—rich in dietary polyphenols—may support mental health by modulating neuroinflammatory responses and the gut–brain axis. Importantly, the results suggest a non-linear association, whereby moderate consumption yields optimal psychological benefit, while excessive intake may not confer further advantages. These findings align with current dietary guidelines that favor polyphenol-rich, plant-based consumption and underscore the need for longitudinal and interventional studies to clarify causality and guide population-specific recommendations [13].

Building on this foundation, the initial aim of this research is to deepen the understanding of the reciprocal impact between dietary patterns and mental well-being, in order to achieve a more precise characterization of the underlying phenomena.

Specifically, we aim to explore whether, within this relationship, the quality of interpersonal relationships may act as a moderating factor that mediates or intensifies this interaction. Based on this objective, we intend to formulate a hypothesis concerning this mediating effect between mental health and nutrition. This approach introduces a relational dimension often overlooked in studies that focuses exclusively on individual variables. However, in light of emerging biopsychosocial perspectives, it may offer valuable insights into the design of comprehensive, culturally sensitive, and socially contextualized intervention strategies. We propose that this perspective avoids the limitations of strict biological approaches and may help foster a more holistic understanding of future interventions promoted by public health organizations.

This review does not merely catalog studies on nutrition or mental health in isolation. Rather, it aims to explore the interconnectedness between dietary patterns, interpersonal relationships, and psychological well-being. The search strategies were specifically designed to capture studies addressing these three domains together, ensuring thematic coherence from the outset. Although often studied separately, these areas share overlapping mechanisms that merit integrated analysis.

## 2. Materials and Methods

This research employed two complementary methodologies: the PRISMA (Preferred Reporting Items for Systematic Reviews and Meta-Analyses) 2020 systematic review and a narrative review. The rationale for this dual approach lies in the fact that the systematic review yielded limited results, insufficient to fulfill the article’s objective of formulating a hypothesis regarding the mediating effect of interpersonal relationship quality on the link between mental health and diet.

For the purpose of this review, the terms were not defined in a strict operational sense but were conceptually delineated to ensure thematic consistency. “Interpersonal relationships” refer broadly to the quality, structure, and function of social ties, including family, peer, and community bonds. “Mental health” encompasses both positive states (such as well-being) and psychological distress (including stress), as addressed in non-clinical populations. “Nutrition” is understood as dietary patterns, food intake behaviors, and the role of specific nutrients or food groups in psychological functioning. These conceptual boundaries guided the search strategy and thematic synthesis.

### 2.1. The Systematic Review

The systematic review was conducted in accordance with the methodological guidelines established by PRISMA 2020 [14,15], which aim to ensure transparency, comprehensiveness, and reproducibility in systematic reviews across disciplinary fields [15]. The process began with the formulation of a clear and well-defined research question, focused on the potential mediating effect of interpersonal relationship quality in the association between diet and mental health. This initial step guided the definition of inclusion and exclusion criteria, the internal protocol structure, and the selection of the most appropriate information sources.

Although PRISMA 2020 recommends registering the protocol in platforms such as PROSPERO (International Prospective Register of Systematic Reviews), it does not consider it mandatory [14] (p. 2), [15] (p. 3). Given the theoretical exploratory nature of this review and the multidisciplinary scope of the relevant evidence, we adopted a methodologically rigorous strategy based on a comprehensive search in four high-impact databases: PubMed (biomedicine and neuroscience), Scopus (broad scientific coverage and bibliometric analysis), Web of Science (multidisciplinary focus and high citation traceability), and APA PsycINFO (psychology, mental health, and interpersonal relationships).

We used the following databases: PubMed (U.S. National Library of Medicine, Bethesda, MD, USA), Scopus (Elsevier, Amsterdam, The Netherlands), Web of Science (Clarivate Analytics, Philadelphia, PA, USA), and APA PsycINFO (American Psychological Association, Washington, DC, USA). The text was prepared using Microsoft® Word for Microsoft 365 MSO (version 2506 build 16.0.18925.20076) 64-bit (Microsoft Corporation, Redmond, WA, USA).

Following this search, a double—blind and independent—screening process was applied to the results, followed by an eligibility assessment based on predefined criteria. Reasons for exclusion were also documented [15]. Data extraction was carried out systematically, covering study design, participants, measures, and relevant outcomes. Given the methodological heterogeneity of the studies included, a structured narrative synthesis was adopted, as recommended by PRISMA in cases where meta-analysis is not appropriate [14] (p. 2). In this case, the narrative synthesis was organized around three axes of analysis: (1) mental health and diet; (2) mental health and interpersonal relationships; (3) diet and interpersonal relationships.

#### 2.1.1. Databases Consulted and Search Strategy

The literature search was conducted between February and 25th March 2025 across three internationally recognized databases: PubMed, Web of Science (WOS), and APA PsycINFO (APA PsycNet). These databases were selected due to their complementary coverage of the biomedical, psychosocial, and behavioral sciences. For each database, a tailored and coherent search strategy (complete strategy can be found in the Appendix A) was developed, structured around three conceptual blocks:Mental health: using the terms mental health, well-being, and stress;Nutrition: using the terms diet, gastronomy, nutrition, food, and nutrients;Quality of interpersonal relationships: using the terms interpersonal relationships, socialization, and conviviality.

In APA PsycINFO, thematic subcategories from the system’s controlled thesaurus were also employed to refine the searches:Category 3314 (Interpersonal Relations), combined with mental health, well-being, and stress;Category 3360 (Health Psychology), combined with social;Category 3370 (Health), combined with mental health services.

#### 2.1.2. Inclusion and Exclusion Criteria

The following inclusion criteria were applied: (a) studies published within the past 20 years; (b) systematic reviews and meta-analyses; (c) studies focused on human populations; (d) publications in English or Spanish.

As an exclusion criterion, all studies exclusively focused on populations with clinical psychiatric or somatic diagnoses were omitted, as the objective was to analyze processes in the general or healthy population within non-clinical settings.

#### 2.1.3. Results Consolidation and Duplicate Control

The search results from three databases were exported and merged into a single matrix. Duplicate entries were then removed through manual screening, based on bibliographic criteria (author, title, year, and journal). Subsequently, a preliminary screening of titles and abstracts was conducted to confirm the relevance of each study in relation to the review’s objectives.

#### 2.1.4. Conceptual Organization and Classification Procedure

To conceptually structure and analyze the retrieved literature, a triangular model was designed, with each vertex representing one of the three central axes: mental health, diet, and quality of interpersonal relationships. Each article was assigned to one or more sides of the triangle based on its thematic focus. Studies that substantially addressed all three axes were placed at the center of the model. The initial classification was performed by one researcher and subsequently reviewed independently by a second researcher. Minor discrepancies were resolved through consensus, ensuring classification reliability and interpretive consistency while minimizing bias.

#### 2.1.5. Data Extraction and Synthesis of Results

The extracted data was organized around key claims derived from the selected articles, thematically grouped according to the three primary axes and their possible interactions. Given the methodological and epistemological diversity of the included studies, a structured narrative synthesis was chosen. This approach allows for a critical interpretation of the findings without imposing a forced quantitative integration.

#### 2.1.6. Methodological Quality of the Included Studies Assessment

Following the Cochrane recommendations, the Joanna Briggs Institute (JBI) Critical Appraisal tool for use Analytical Cross-Sectional Studies [16]; Systematic Reviews [17]; Textual Evidence: Policy [18]; Cohort Studies [16]; Randomized Controlled Trials [19]; Textual Evidence: Narrative [18]; Quasi-Experimental Studies [20] was used for methodological quality (risk of bias) study assessment.

### 2.2. The Narrative Review

The PRISMA-guided systematic review revealed a significant scarcity of empirical studies examining the mediating effect of interpersonal relationship quality between mental health and nutrition. Moreover, the few existing studies address this relationship largely at a descriptive level, without offering a theoretical or logical framework capable of explaining the mediating role of interpersonal relationships. For this reason, employing a narrative review is both methodologically justified and scientifically appropriate. This type of review not only allows for the integration of findings from diverse disciplines but also helps identify knowledge gaps and generate plausible hypotheses to guide future research [21]. Unlike systematic approaches—which require narrowly defined questions and strict inclusion criteria—narrative reviews offer a flexible, though not arbitrary, framework for exploring complex constructs that remain insufficiently examined through empirical means [22]. In fact, when the available literature is fragmented, preliminary, or methodologically heterogeneous, the narrative review provides suitable interpretive tools to synthesize findings critically and generate emerging conceptual models [23]. Nonetheless, this flexibility requires an explicit commitment to transparency, scope delimitation, and methodological justification at each stage [24].

The narrative review follows a rigorous methodological sequence:Define a focused and relevant topic with a clearly identified audience, avoiding overly broad approaches [22]. In our case, the research question centers on identifying the underlying rationale that explains the mediating effect of interpersonal relationships between diet and mental health, with the aim of enabling a future empirical investigation to refine this effect.Design a deliberate and justified search strategy—though not necessarily systematic—appropriate to the field and its objectives [23].

In our case, the information search was multidisciplinary, drawing from the following:Anthropology [24,25,26,27];Esthetics [28,29];Psychology [30];History [31,32,33];Ethics [34];Urban architecture [35];Gastronomy [36];Culinary arts [37,38,39,40].

This corpus required the inclusion of both academic and professional literature, including influential non-academic authors whose practical experience and public recognition make them key references in the culinary arts. A guiding selection criterion was that these authors reflect critically on the interconnections between the three central axes—mental health, diet, and interpersonal relationships—offering conceptual frameworks to articulate these links.
Conduct a critical reading of the selected material, allowing for reflective adjustments in focus as the process unfolds [23].Organize findings into a coherent structure—whether thematic, chronological, or conceptual—to facilitate interpretation [22].Develop an interpretive synthesis that identifies patterns, tensions, or gaps and contributes to theoretical development [21].

Steps (1) and (2) have been outlined above; step (3) is addressed in the Results section, while steps (4) and (5) are discussed in the Discussion section.

## 3. Results

### 3.1. Results of the Systematic Review

Initially, 327 articles were found from the various databases consulted. When duplicates were eliminated, the remaining number was 276. Articles were subsequently eliminated based on the exclusion criteria indicated. However, we found several articles that only touched on some of the three topics (nutrients, mental health, and interpersonal relationships) without offering direct clues about the interaction. They merely mentioned the interaction, citing other research. In the end, we found only 23 articles that were useful for the research. For more details, see Figure 1.

The characteristics of the included studies can be found in Appendix A.

The final selection encompassed a diverse set of 23 articles spanning seven methodological categories, ensuring a comprehensive view of the relationship between diet, mental health, and interpersonal relationships. Specifically, the review included the following:

Analytical cross-sectional studies (*n* = 7), which provided observational data on associations between variables such as food insecurity, psychological distress, and social support in diverse populations.

Systematic reviews (*n* = 4), synthesizing existing evidence on the effects of nutrition and lifestyle interventions on mental and emotional well-being.

Textual evidence from policy documents (*n* = 3) and narrative sources (*n* = 2), contributing conceptual and contextual insights into the integration of nutrition and mental health services in public health and clinical settings.

Cohort studies (*n* = 3), offering longitudinal perspectives on the interplay between early-life nutrition, mental health, and long-term outcomes.

Randomized controlled trials (*n* = 3), which tested the efficacy of interventions such as wellness programs, diet-based treatments, and market coupon initiatives on psychological and nutritional outcomes.

Quasi-experimental studies (*n* = 1), addressing behavioral change and health perception through digital nutrition resources.

This heterogeneity in study design allowed for a nuanced, multi-angle synthesis of the evidence, supporting robust conclusions on the interdependence of diet, mental health, and social context.

The full study on the quality of the studies can be seen in Appendix A.

All 23 included articles were critically appraised using the appropriate JBI checklists according to study design. The overall methodological quality was satisfactory, with most studies meeting the majority of the JBI criteria.

Among the analytical cross-sectional studies (*n* = 7), five scored high, fulfilling more than 80% of the checklist items, while two presented moderate risk of bias due to unclear confounding strategies or incomplete reporting on outcome measurements.

The systematic reviews (*n* = 4) showed robust quality, all meeting core standards for methodological rigor, search strategy transparency, and critical appraisal of included sources.

The randomized controlled trials (*n* = 3) were generally well conducted, although one study lacked detailed reporting on blinding procedures and allocation concealment, reducing its overall appraisal rating.

The cohort studies (*n* = 3) showed variation in follow-up completeness and confounder control, with two rated as high quality and one as moderate.

The quasi-experimental study (*n* = 1) met most criteria but showed limited clarity on outcome measurement reliability.

Textual evidence articles (policy and narrative, *n* = 5) were appraised for credibility, coherence, and relevance; all were considered valid sources for conceptual enrichment, though not subject to empirical validation.

In sum, the body of evidence was deemed sufficiently robust to support the review’s synthesis, with limitations transparently considered in the interpretation of results.

The classification of the different articles according to the axes of study is presented in Table 1 below.

The relationship between nutrition and mental health has traditionally been examined from isolated biomedical or psychological perspectives. However, in recent decades, a more holistic approach has emerged, one that acknowledges the interaction between diet and psychological well-being, as well as the potential moderating role that interpersonal relationships may play in this dynamic, as will be discussed below.

This study proposes a triangular framework in which each side represents the relationship between two of the key variables (nutrition/diet, mental health, interpersonal relationships), while the center of the triangle symbolizes their complex and synergistic intersection.

All the articles reviewed addressed the three core themes: mental health, interpersonal relationships, and nutrition. However, for the sake of greater analytical clarity and to deepen the understanding of how these dimensions interact, we have chosen to present the findings according to thematic pairs (nutrition and mental health, relationships and mental health, nutrition and relationships). This structure allows for a more detailed exploration of each connection, gradually contributing to a more integrated and comprehensive picture of the phenomenon under study.

Accordingly, the results of the systematic review are organized as follows:

#### 3.1.1. Axis 1: Nutrition and Mental Health

The systematic review confirms a direct relationship between nutrition and mental health, underscoring the importance of approaching diet as a decisive factor in psychological well-being. Dietary habits influence emotional states through the regulation of neuroinflammatory, hormonal, and neurochemical mechanisms. For instance, stress can trigger inflammatory responses that are amplified by poor dietary patterns, creating a negative feedback loop that exacerbates psychological distress [41]. Within this framework, the ketogenic diet has been the subject of recent studies exploring its use as an adjunct therapy for psychiatric disorders such as depression and schizophrenia, although further research using more rigorous methodologies is required to evaluate its effectiveness [42].

Several studies included in the systematic review adopt holistic approaches that integrate physical, psychological, and social dimensions [43], showing that the combination of a balanced diet, sufficient sleep, and regular physical activity correlates positively with self-perceived health. Complementary research suggests that integrated lifestyle interventions among individuals with mental health disorders can significantly reduce metabolic risk factors and promote general well-being [44].

The review also highlights a paradigm shift in the understanding of the diet–well-being relationship. Instead of prioritizing weight loss as the primary goal, some authors advocated for a model of intuitive and non-restrictive eating that emphasizes metabolic health and emotional balance [45,46]. Additionally, one study examines how dietary and physical activity changes linked to remote work negatively impact emotional balance, offering several strategies for remote interventions [46].

From a human development perspective, multiple studies have shown that nutritional interventions during critical life stages—such as pregnancy and early childhood—have long-term implications for cognitive and emotional health. Several investigations indicate that appropriate prenatal and early childhood nutrition is associated with improved mental health outcomes later in life. In this context, eating ceases to be a purely physiological act and becomes a process with profound psychosocial implications [47,48,58].

Food insecurity is identified as a structural determinant of vulnerability in both developed and developing contexts. In African countries, for example, limited and unstable access to food particularly affects female-headed households, with significant negative effects on mental health [49]. One analysis emphasizes that food security cannot be separated from water security, as access to clean drinking water directly influences nutritional quality and community well-being [50]. Similarly, multiple articles explore how food precarity in both urban and rural settings of developed countries significantly impacts emotional stability, especially among marginalized communities [51,52,53,54,55,56].

In addition, one study introduces a sociocultural perspective, showing that negative self-stereotyping and racial stigma affect body image and dietary behaviors among adolescents from ethnic minority groups, thus establishing a clear link between identity, nutrition, and mental health [57].

#### 3.1.2. Axis 2: Nutrition and Interpersonal Relationships

Nutrition must be understood not only as a biological process but also as a culturally and socially mediated practice. Interpersonal relationships influence dietary decisions, while the social environment can either support or hinder the adoption of healthy eating habits [47,48,49,52,54,55,56,58,59,60]. Several studies show that nutritional interventions in vulnerable households are more effective when integrated with community services that promote autonomy, dignity, and social learning, which in turn have a positive effect on mental health [55,56].

In the same direction, a study on farmers’ markets in Canada offers an interesting example of nutrition-focused intervention spaces [51]. The study finds that these community-based purchasing environments foster a sense of belonging and self-esteem, facilitating the adoption of healthier eating habits. Such forms of communal eating and free food choice can strengthen social bonds and help combat the social isolation experienced by participants in food assistance programs.

Negative self-stereotyping and low self-esteem also influence dietary quality and body mass index, particularly among stigmatized ethnic minorities. One study emphasizes the psychosocial dimension of ethnic identity, showing how belonging to a socially marginalized group can significantly affect dietary patterns [57]. Another study examines how social isolation among young people with compulsive video gaming habits negatively impacts both physical health and dietary behavior [61].

One article analyzed the recovery of body weight following weight-loss programs, comparing a pre-packaged meal plan to one using conventional foods, both supported by behavioral counseling. The findings highlight the importance of behavioral and motivational factors in sustaining dietary changes [62]. Two studies evaluate the effectiveness of nutritional and lifestyle interventions when delivered in supportive community or outpatient settings [44,45].

Changes in the world of work—particularly the rise in remote work during the COVID-19 pandemic—have significantly affected relational dynamics and lifestyle habits. One reviewed study examines how reduced physical activity and shifts in dietary patterns associated with home-based telework negatively impact overall well-being [46]. The study assesses the effectiveness of telehealth interventions aimed at promoting healthy habits—both behavioral and nutritional—as a strategy to mitigate the adverse effects of prolonged sedentarism and dietary disruption in domestic work environments.

#### 3.1.3. Axis 3: Mental Health and Interpersonal Relationships

Social relationships are critical determinants of mental health. One article examined how self-financing programs targeting women not only improve child nutrition but also enhance female empowerment, self-esteem, and psychological well-being [59]. Several articles analyze interventions aimed at improving psychological well-being, highlighting the role of support from healthcare professionals as a key factor in increasing the effectiveness of such interventions [44,45]. In this same line, another study shows that healthcare accompaniment in the implementation of nutritional or mental health protocols for pregnant women can reduce the risk of mental health problems such as anxiety or depression [47].

One article underscores how job insecurity and racial discrimination negatively affect the mental health and job performance of migrant workers [63]. Specifically, ethnic discrimination contributes to negative self-perception that increases psychological risk factors, unless moderated by a high degree of self-esteem [57].

Multiple studies converge on the finding that economic insecurity—and consequently, food insecurity—and the absence of support networks significantly increase the risk of depression, anxiety, and social isolation [51,52,53,55,56,60].

Finally, one study provides evidence that early psychosocial stimulation has a greater long-term impact on emotional well-being than nutrition alone. This finding reinforces the importance of considering the relational dimension in early intervention strategies [58].

#### 3.1.4. Common to All Axes: The Intersection of Nutrition, Mental Health, and Interpersonal Relationships

At the point of convergence among the three dimensions analyzed—nutrition, mental health, and interpersonal relationships—several key patterns emerge:Food insecurity functions as a transversal axis linking malnutrition, psychological stress, and social exclusion [49,52,53,54,56];Interpersonal and community networks play a protective role, helping to buffer or modulate the negative effects of structural determinants [47,51,53,54,57,60];There is a clear need for integrated intervention strategies that simultaneously address nutritional education, the strengthening of social and community support networks, and comprehensive mental health promotion [44,45,46,51,55,58].

#### 3.1.5. Gaps in the Systematic Review

The studies included in the systematic review were insufficient in design or methodology to adequately address the research question, which aimed to conceptually understand the articulation of the three core domains: mental health, nutrition, and the quality of interpersonal relationships. Regarding the mediating or moderating role of interpersonal relationships in the connection between nutrition and mental health, the reviewed articles provide only partial insights. Several significant knowledge gaps remain that merit further investigation.

First, there is a lack of research explicitly examining the quality of interpersonal relationships as a mediator. Current studies often refer to social dimensions in structural terms—such as poverty, community, belonging, or discrimination—but do not measure the affective or functional quality of close personal ties (e.g., friendships, partnerships, family bonds, caregiving networks) or evaluate whether such relationships modulate the effects of poor or healthy nutrition on mental health or stress [49,50,51,52,53,54,55,56,57,58,59].

Second, there is little to no exploration of social relationships as moderators of physiological effects of nutrition. It remains unclear whether high-quality relational environments buffer pathophysiological processes associated with poor diet. It would be relevant to examine whether an individual with a deficient diet but strong, warm interpersonal relationships show less psychological deterioration compared to someone under similar nutritional conditions but living in isolation or conflictual relationships. One article suggests an interaction between diet and emotion but does not address the role of relational quality [41].

Third, the role of family and close interpersonal relationships in adult dietary behavior is underexplored. While many studies focus on children or community-level interventions, few examine how emotional support or affective co-regulation in adult relationships (e.g., with partners, family, or friends) influences adherence to healthy diets and impacts stress and well-being. Some studies contrast dietary approaches with body acceptance models, but do not account for the interpersonal context in which such choices are made [45].

Fourth, relational interventions that address both dietary habits and psychological well-being have not been studied. While some articles explore community-based or dietary programs, they rarely assess whether strengthening affective bonds—rather than general community connectedness—produces dual benefits in nutrition and mental health. The notion of “sense of community” is mentioned, but without distinguishing generic social networks from the quality of meaningful personal [51,52,53,54,55].

Fifth, there is a lack of longitudinal studies analyzing how relational, dietary, and psychological trajectories evolve over time. No studies were found that track changes in mental health in parallel with changes in diet and relational quality. There is a need for longitudinal mediation or moderation models that account for multiple variables—diet, relationships, life events, social support, and emotional health. While one longitudinal study was identified, it does not explicitly cross-reference relational quality with nutritional trajectory and mental health outcomes [58].

Sixth, few studies apply validated instruments for measuring relational quality in the context of diet and mental health. Almost none use standardized tools such as *The social support questionnaire* [64] or the *Quality-of-life-inventory* [65], among others. As a result, no robust quantitative analyses are available to determine whether relational quality enhances, neutralizes, or worsens the link between diet and psychological well-being.

All points of the checklist (Appendix A) were verified to ensure that all methodological requirements included in PRISMA 2020 have been met.

### 3.2. Results of the Narrative Review

In his 1860 work *The Physiology of Taste*, Brillat-Savarin (2010) emphasizes the need for a reasoned understanding of all that pertain to human beings and their relationship with food. He advocates for a comprehensive form of knowledge that integrates multiple disciplines. His fourth aphorism—“Tell me what you eat, and I will tell you what you are” [36] (p. 15)—invites reflection on human nature and its connection to nourishment. Similarly, Kass (2005), in *The Hungry Soul*, calls for a multidisciplinary approach, viewing food as a bridge between scientific and humanistic ways of understanding. In the act of eating, the human being is invited into a path of virtue and conscious choice. Our human nature—as a unified psychophysical being endowed with spirit and a sense of meaning—positions us as open to relationships with others and the world [34].

In this narrative review, we focus on anthropologists, philosophers working in esthetics, historians, psychologists, and architects who have made explicit contributions to the field of food and nourishment from within their disciplines. For the same reason, we also include culinary professionals who reflect philosophically on their craft, linking it to anthropology, esthetics, and psychology.

#### 3.2.1. Anthropology

From a philosophical and anthropological perspective, Cruz-Cruz (1991) argues that the relationship between human beings and food must be understood holistically, avoiding any analytical fragmentation of the cultural context in which this relationship is embedded. Within this integrative framework, food is not merely a biological commodity, but also a vehicle of symbolic meaning that reflects intellectual, moral, and social dimensions of human existence. These meanings transcend the physiological function of eating and reveal constitutive aspects of human nature, such as corporeality, psyche, sociality, and the capacity to confer meaning through shared symbols [24].

Cruz-Cruz conceptualizes food as a “nutritive product, desired product, and customary product” [7] (p. 16), emphasizing its triple nature—biological, sensory, and cultural. This view allows us to understand food as a mechanism of social cohesion, expressed through customs, rituals, and shared practices. As he states, “shared meals, even simple ones, render human behavior more spiritual and social” [24] (p. 26), recognizing the dignity of both self and other through the act of eating together.

In a complementary approach, Harris (2009) examines dietary habits through the lens of cultural materialism, arguing that food choices are determined by ecological, climatic, economic, and resource availability conditions. From this perspective, the human diet is shaped not only by nutritional considerations but also by ecological efficiency and economic value. As Harris notes, “what is good to eat” is often “what is good to sell,” regardless of its actual nutritional properties [27] (p. 18). Thus, dietary choices reflect both adaptive responses to the environment and culturally constructed preferences, going beyond mere taste or nutritional value.

Finally, Dobre (2021) emphasizes that food engages human beings on multiple levels. Eating connects with emotions, memories, affective bonds, and shared experiences, functioning as language, symbol, and history. In this sense, food reveals how individuals position themselves in the world and relate to others. However, the author warns of the risk of depersonalization posed by the rise in the food industry, which increasingly replaces traditional caregiving roles—such as the mother or grandmother—thereby eroding the affective and symbolic dimensions historically associated with the act of nourishing [25].

#### 3.2.2. Esthetics

Various philosophical and sociocultural approaches have enriched the contemporary understanding of food, emphasizing its cognitive, symbolic, and relational dimensions. Korsmeyer (1999) argues that the appreciation of food should not be understood solely as a sensory response, but as a vehicle of meaning shaped by cultural, social, and personal factors. Food preferences—such as the fondness for spicy flavors in Mexico or the avoidance of pork in Jewish traditions—function as expressions of collective identity and symbolic belonging. In this sense, the experience of eating articulates a subject’s relationship with themselves and the world they inhabit, integrating embodiment, culture, and consciousness [29].

From a contemporary philosophical standpoint, Jaques and Vilar (2024) assert that food constitutes a foundational cultural fact in all forms of organized life. For these authors, “eating with meaning” [28] (p. 52) implies recognizing that every act of eating is simultaneously biological and symbolic: it is not merely a matter of nutrient intake but of participating in a network of meanings. Eating and cooking, beyond their utilitarian function, emerge as second-order reflective practices that enable critical distance from established dietary habits. This opens the possibility of reshaping one’s life through conscious eating, paving the way for ethical and social transformation.

Le Breton (2007), through an esthetic and anthropological lens, warns of the homogenization of taste among younger generations, influenced by standardized food habits of North American origin. According to the author, the widespread consumption of ultra-processed foods—rich in fats and sugars—diminishes the sensory and symbolic richness of taste, leading to a form of cultural impoverishment. Le Breton emphasizes that taste formation is a complex phenomenon involving both biological data and educational processes, but it is ultimately the symbolic dimension that prevails. Culture shapes the palate beyond biology. Consequently, food preferences are not merely individual choices but socially constructed acts that reflect the symbolic structure of the community to which one belongs [26].

#### 3.2.3. History

According to Perlès (2011), food has always gone beyond its primary function of sustenance, integrating a complex system of signs that organizes human coexistence, belonging, and conceptual thinking. Since its origins, humanity has experienced conviviality through the act of eating, reinforcing community bonds and articulating cultural codes [33].

Freedman (2021) emphasizes that the study of food cannot be reduced to health data or environmental conditions, since human dietary behavior is often driven not by scientific evidence but by emotions, resentments, lived experiences, and other non-material factors. The author points out that variables such as ethnicity, gender, and socioeconomic status significantly influence food choices [32].

Flandrin and Montanari (2011) expand on this perspective by arguing that the dining table and eating habits are privileged spaces for the expression of cultural particularities, national identities, and even religious disputes. The rules that govern these spaces are not neutral; they respond both to identity differentiation processes and to a mysterious or arcane perception of food—understood not only as a symbol, but as a real vehicle for transmitting values and qualities. Within this framework, eating also entails materially appropriating deeply embedded cultural meanings [31].

In contemporary European contexts, the social function of eating remains essential. Meals are conceived not merely as moments of nutritional intake but as social rituals where family and emotional bonds are reaffirmed, and where gastronomic pleasure is interwoven with the human need to share [31]. This convivial pleasure depends on shared time and is often expressed through a degree of ceremony—even in informal settings, such as sharing tapas with friends. These forms of sociability stand in contrast to more functional and depersonalized modes of food consumption, such as eating processed foods alone in front of the television or during mass events, where the relational dimension of eating becomes diluted.

#### 3.2.4. Psychology

Several studies in the fields of sensory psychology and gastronomy have emphasized the significant influence of the social context on eating. Spence and Piqueras-Fiszman (2014) highlight that social interaction can enhance the enjoyment of a meal, though they note that this effect largely depends on the quality of the company. Beyond the hedonic component, the presence of other diners significantly impacts not only the types of food consumed but also the quantity. Individuals tend to mimic the food choices and portion sizes of those around them—a phenomenon of social mimicry that shapes eating behavior. Moreover, it has been observed that the duration of time spent at the table increases in proportion to the number of people present, which in turn can influence the overall amount of food consumed [30].

#### 3.2.5. Urban Architecture

In *Ciudades hambrientas*, Carolyn Steel (2020) presents food as an essential organizing axis through which various disciplines involved in urban development can engage in dialog about sustainable city models [35]. According to the author, shared meals represent one of the most powerful social bonds, as eating together involves accepting shared norms of encounter and belonging. In this sense, the act of eating transcends its biological dimension to become a tool of social organization—capable of inclusion or exclusion and even functioning as a mechanism of control [35].

Steel argues that food practices directly influence the formation of community ties. Sharing food promotes a sense of proximity, while cultural differences related to eating can create symbolic distance. She also emphasizes that the loss of intergenerational commensality has a negative impact on children’s social development. In the British context, several studies have linked the absence of family meals with increased mental health issues among the youth, including depression, anxiety, and attention disorders—conditions associated with a lack of essential relational skills typically acquired in the domestic setting [35].

The author further critiques the growing individualization of eating, particularly during childhood. She highlights how the food industry has promoted highly processed products specifically targeted at children—softer, sweeter, and less nutritious than those intended for adults—describing this as a reckless and counterproductive strategy for healthy habit formation. In her view, such practices have contributed to the reinforcement of dietary patterns that exacerbate the obesity crisis in developed countries. Steel asserts that this phenomenon cannot be addressed solely through a nutritional lens: obesity is, ultimately, a symptom of deeper dysfunctions in contemporary lifestyles. Consequently, she calls for a comprehensive rethinking of food culture—one that integrates its urban, social, and symbolic dimensions [35].

#### 3.2.6. Culinary Arts

In *Cocinar, comer, convivir*, chef Andoni Luis Aduriz and philosopher Daniel Innerarity [37] argue that food is far from a purely biological act—it is a profoundly symbolic and socially significant practice. They maintain that eating is not simply about satisfying a physiological need but rather involves a complex network of meanings that challenge our understanding of society, collective organization, and well-being. In a context increasingly dominated by functionalist and instrumental discourses around food, the authors call for a more reflective perspective—one in which eating is seen as a manifestation of the art of living. From this standpoint, what they term “food illiteracy” [37] (p. 181) is not merely a lack of nutritional knowledge, but a form of cultural disconnection that can contribute to disordered eating and social alienation.

This relational dimension of food is also emphasized by other voices in the culinary world. Chef María Nicolau (2023) asserts that food is a medium through which we relate to the world, to others, and to ourselves. It has the power to transform and shape reality through the relationships it generates [40]. Similarly, chef Massimo Bottura, through his *Refettorio* initiative launched in 2015, has proposed a socially transformative model of gastronomy. His project seeks to restore the dignity of individuals in vulnerable situations by offering them the experience of fine dining. Guests are treated as restaurant patrons, which helps rebuild their self-esteem and sense of belonging [39].

In the same humanistic spirit, chef José Andrés (2021), through his organization World Central Kitchen, has delivered emergency food relief in crisis zones—including armed conflicts, natural disasters, and the COVID-19 pandemic. His approach not only addresses nutritional needs but also provides emotional comfort by serving meals adapted to the cultural norms of the affected populations. For Andrés, food is a tool of holistic care, stating that “cooking has the power to build bridges” [38] (par. 4).

The thematic organization of these findings, the formulation of the central hypothesis, and their connection to the systematic review are addressed in the following Discussion section.

## 4. Discussion

In this section, we offer a conceptual framework for the entire narrative review, following the same structure used in the systematic review. This parallel organization allows for meaningful comparison between the two. We then relate the findings from the narrative review to those of the systematic review. Finally, we conclude by presenting our proposed hypothesis.

The reviewed literature reflects a growing concern with understanding food as a multidimensional phenomenon—one that integrates biological, symbolic, and relational dimensions. Based on this premise, we organize the contributions of the various authors according to the axis of the triangle that predominates in their analysis.

### 4.1. Organization of Narrative Review Results According to the Structure of the Systematic Review

The following Table 2 presents the thematic organization of the findings emerging from the narrative review. The articles are grouped according to the predefined conceptual axes, with the aim of providing a structured overview and facilitating the identification of emerging patterns and interrelations. The order is as follows:

#### 4.1.1. Axis 1: Nutrition and Mental Health

Although mental health is not the central focus of all the works reviewed, some authors implicitly address the relationship between diet and psychological well-being through symbolic, affective, or cultural lenses [26,27,28,29,30,31,32,33,34,35]. These authors emphasize how contemporary changes in dietary habits—shaped by cultural and social norms—affect psychological well-being, especially among younger generations. Along similar lines, Dobre (2021) warns that the food industry has replaced traditional caregivers, such as mothers or grandmothers, which may lead to a risk of depersonalization and weaken the emotional bond with food [25].

#### 4.1.2. Axis 2: Nutrition and Interpersonal Relationships

This axis includes contributions from multiple authors who highlight the relational dimension of food, understood as a cultural practice, a bearer of meaning, and a promoter of social ties [24,25,26,27,28,29,32,37,38,39,40].

Historians also align with this axis, asserting that dietary norms transform the shared dining experience into a privileged space for the expression of collective identity [31,32,33].

#### 4.1.3. Axis 3: Mental Health and Interpersonal Relationships

This axis features two books from the narrative review—Spence and Piqueras-Fiszman (2014) as well as Steel (2020)—who explore how the social context of eating can directly influence individuals’ psychological states [30,35].

#### 4.1.4. Common to All Axes: Intersection of Nutrition, Mental Health, and Interpersonal Relationships

An integrative synthesis of all three axes can be found in the works of several authors [24,25,26,27,28,29,30,31,32,33,34,35,36,37]. One author proposes that the human need to eat offers a pathway for individuals to choose how they wish to be in the world [34]. Two authors articulate the biological, moral, and symbolic aspects of food, thus integrating health, culture, and social life [24,25,26,27,28,29,30,31,32,33,34,35,36,37]. Another contribution condenses this perspective by offering a comprehensive reading of food culture—one that encompasses its impact on mental health, urban dynamics, and forms of sociality [35].

### 4.2. Comparison Between the Narrative and Systematic Reviews

#### 4.2.1. Convergences Between the Narrative and Systematic Reviews

Although the two reviews are grounded in different methodologies—the first being empirical and systematic, the second being theoretical and humanistic—that both converge in proposing a broader understanding of food as a multidimensional phenomenon. They articulate the relationship between nutrition, mental health, and interpersonal relationships through a holistic framework. This shared perspective makes it possible to organize contributions along the three axes and their synergistic intersection.

Regarding the first axis (nutrition and mental health), both reviews affirm that nutrition directly influences psychological well-being [25,26,41,42].

With respect to the second axis (nutrition and interpersonal relationships), both reviews robustly underscore the relational dimension of food and emphasize that this dimension is inseparable from the surrounding social context. The loss of shared food spaces can weaken cultural and social bonds [24,27,28,31,33,51,55,57].

Concerning the third axis (mental health and interpersonal relationships), both approaches agree that social context significantly modulates psychological well-being [30,35,54,56,60].

At the intersection of all three axes—nutrition, mental health, and interpersonal relationships—both reviews converge on the insight that a comprehensive understanding of eating requires the integration of these three dimensions [24,34,36,38,45,46,58].

#### 4.2.2. Contribution of the Narrative Review

We found that while empirical research has demonstrated the influence of the three study themes (nutrients, interpersonal relationships, and mental health), it has not been able to fully offer a narrative that unifies the three themes or articulate an explanation of the interaction between them. The narrative review complements the systematic review by offering a deeper, symbolically rich, and culturally contextualized understanding of human food-related behavior. It incorporates disciplines traditionally excluded from biomedical or psychosocial analyses, such as philosophy, anthropology, esthetics, history, and architecture.

The narrative review does not replace scientific evidence but expands and enriches it through interpretive tools rooted in the humanities. It offers a more complex, critical, and situated perspective on food, illuminating dimensions often rendered invisible by exclusively empirical models. This perspective opens the door to innovative approaches in food-related interventions, grounded in ethics of care, relationality, and cultural awareness.

A common denominator throughout the narrative review seems to be the holistic nature of food, which intersects highly personal themes such as the expression of the individual’s personal identity and that of the society to which they belong, as well as the expression of the significance of this world, and of ethical and cultural considerations. They also see its educational value as a way of educating society and point out that all this meaning outweighs the hedonistic value of food, or perhaps it could be said that it is the personal and social significance that gives food its hedonistic value. Completing this view from the culinary arts, creative expression is highlighted as the desire to bring out the best in oneself, even as a space for self-regeneration through one’s actions.

In summary, we could say that food must be understood within the “symbol” category, which has been explicitly mentioned by many. The symbol category would indicate that there is a thing (objective element) that carries a meaning (subjective element), which is what gives it value.

### 4.3. Hypothesis Formulation

Based on the findings of this study, we propose the following hypothesis: the quality of interpersonal relationships acts as a catalytic and amplifying factor in the effect of nutrition on mental health. If supported, this hypothesis would offer an explanation for the interindividual variability in the influence of an objectively measurable reality—namely, nutrients—on a reality that is both objective and subjective: mental health.

Human nutrition rests on an objective foundation—nutrients—whose composition and quality can be empirically measured. However, as this research has demonstrated, the impact of nutrition on mental health does not depend solely on biological variables. Rather, it is modulated by subjective and contextual dimensions.

Mental health, as a biopsychosocial phenomenon, includes objective components—such as clinical and functional indicators—as well as a subjective dimension that involves individual perception, emotional experience, and environmental context. Within this framework, we propose that the quality of interpersonal relationships is a key explanatory variable in accounting for the variability observed in the link between dietary patterns and mental health outcomes.

Considering the research, we offer a rationale for the hypothesis that connects the most subjective with the biological perspective:

The symbolic nature of food makes it a carrier of meaning that highlights the quality of interpersonal relationships. This has been demonstrated by the narrative review. Furthermore, it is well known that the quality of interpersonal relationships is the reality that has the greatest influence on human beings in terms of stress management and resilience processes [66] (pp. 279–298). This would lead to putting the body in the best position to maximize the effect of nutrients in this context, as shown by the systematic review.

Functionally, we suggest that interpersonal relationships may serve a catalytic role by facilitating or accelerating psycho-emotional processes that enhance the positive effects of nutritional interventions. This may explain why nutrients do not produce uniform or immediate effects on mental health across individuals. At the same time, relationships may play an amplifying role, magnifying the beneficial impact of nutrition on mental well-being. Together, these functions suggest that the interaction between diet and mental health cannot be fully understood without incorporating the relational dimension as an essential mediating or moderating factor.

### 4.4. Future Directions

In the context of interventions aimed at improving mental health through nutrition, it is essential to incorporate a systematic assessment of the quality of interpersonal relationships. In addition to identifying the full nutritional profile, the participant’s mental health status, and individual characteristics, this assessment should consider not only the healthcare environment—including the quality of the therapeutic alliance—but also the participant’s family, social, and occupational contexts, given their potential influence on the effectiveness of dietary strategies. In other words, the quality of interpersonal relationships should be evaluated both in those directly involved in the intervention and in the broader network of individuals who regularly interact with the participant.

Based on this premise, we recommend structuring future research around four key variables:The participant’s profile;The prescribed dietary pattern;The nature of the mental health issue;The quality of the participant’s interpersonal relationships.

The study sample could be organized into two primary groups based on interpersonal relational quality (high vs. low), using a combination of validated indicators for social support and relational quality. Each group would then be subdivided into two experimental conditions: one receiving a specific dietary intervention and the other a placebo. In this context, the placebo effect deserves special attention, as the outcomes involve highly subjective processes.

This would yield four experimental groups:High Interpersonal Quality (HIQ) + Nutrients (N);HIQ + Placebo (P);Low Interpersonal Quality (LIQ) + Nutrients (N);LIQ + Placebo (P).

This design would allow for a comparative analysis of whether improvements in mental health vary based on both dietary intervention and interpersonal support.
The comparison between HIQ+N and LIQ+N would assess the catalytic and amplifying effect of interpersonal relationship quality.The comparisons between HIQ+N vs. HIQ+P and LIQ+N vs. LIQ+P would evaluate the placebo effect.

If the results show that participants with low interpersonal relational quality do not experience significant improvements in mental well-being despite nutritional intervention while those with strong interpersonal networks do, this would offer empirical support for the hypothesis that interpersonal relationships serve as a critical moderator or enhancer of the impact of nutrition on mental health.

## 5. Conclusions

This study has explored the relationship between nutrition and mental health through the mediating lens of interpersonal relationships, combining findings from both a systematic review of empirical literature and a narrative review grounded in the humanities. The convergence of both approaches supports the hypothesis that interpersonal relationship quality plays a key role in modulating and enhancing the impact of nutrition on psychological well-being.

While existing research has extensively documented the biological and psychological pathways linking diet to mental health, this study identifies a significant gap in understanding the relational mechanisms involved. The narrative review adds interpretive depth by highlighting how cultural, symbolic, and affective dimensions of food—often overlooked in biomedical models—can reshape the way individuals experience eating, healing, and belonging.

The proposed hypothesis—that interpersonal relationships serve as a catalytic and amplifying factor in the diet–mental health dynamic—opens a promising path for future interdisciplinary research.

The rationale for this hypothesis lies in the fact that the symbolic nature of food makes it a carrier of meaning that reflects the quality of interpersonal relationships, which, in turn, makes it the most influential factor in stress management and resilience processes. This would promote better body state, thus enhancing the effect of nutrients.

Testing this hypothesis through controlled experimental designs could lead to more holistic, context-sensitive interventions in public health and clinical nutrition. Such approaches would not only improve efficacy but also promote ethical care practices rooted in relationality, cultural awareness, and human dignity.

In sum, addressing mental health through nutrition requires more than targeting nutrients; it demands attention to the social and symbolic conditions in which eating takes place. Integrating these dimensions can enrich our understanding of health and well-being, ultimately contributing to more comprehensive and humane models of care. 

## Figures and Tables

**Figure 1 nutrients-17-02318-f001:**
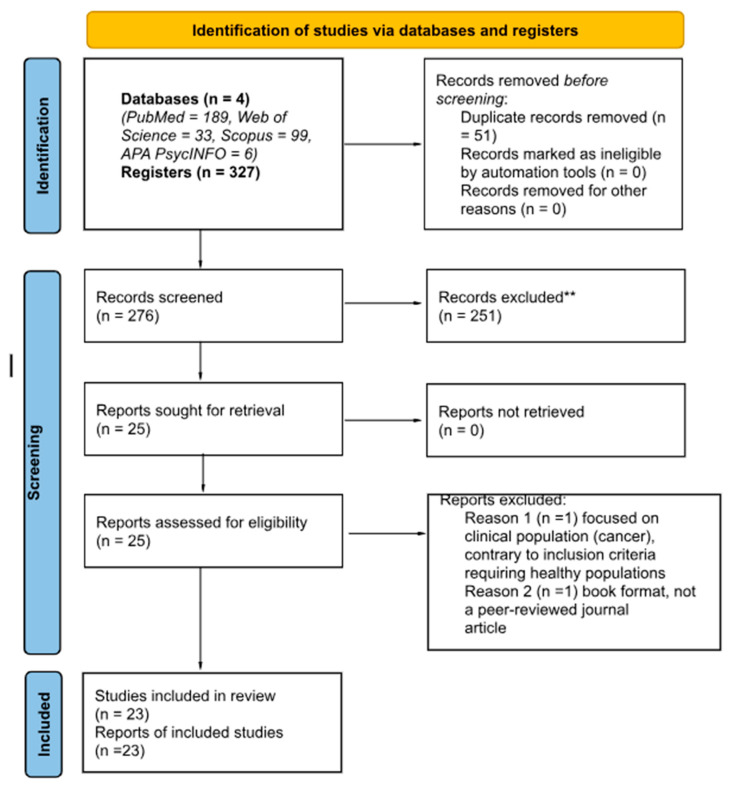
Flow diagram according to [15]. **: In the article exclusion process, no automation tools were used.

**Table 1 nutrients-17-02318-t001:** Relationship between thematic axes and articles from the systematic review.

Thematic Axes	Articles from the Systematic Review
Nutrition—Mental Health	[41,42,43,44,45,46,47,48,49,50,51,52,53,54,55,56,57]
Nutrition—Interpersonal Relationships	[44,45,46,47,48,49,51,52,54,55,56,57,58,59,60,61,62]
Mental Health—Interpersonal Relationships	[44,45,47,51,52,55,56,57,58,59,60,63]
Clear Overlap of All Three Axes	[44,45,46,47,49,51,52,53,54,55,56,57,58,60]

**Table 2 nutrients-17-02318-t002:** Relationship Between thematic axes and articles from the narrative review.

Thematic Axes	Articles from the Narrative Review
Nutrition—Mental Health	[25,26,34,35,36]
Nutrition—Interpersonal Relationships	[24,25,26,27,28,29,31,32,33,34,36,37,38,39,40]
Mental Health—Interpersonal Relationships	[30,34,35,36]
Clear Overlap of All Three Axes	[24,34,36,37]

## Data Availability

The data supporting the findings of this study—specifically, the list of included articles and the results of the systematic search and selection process—are provided in the Appendix A.

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
