# Peer review of "Systematic and Narrative Review of the Mediating Role of Personal Relationships Between Mental Health and Nutrition"

_nutrients, 2025, doi:10.3390/nu17142318_

Round 1

Reviewer 1 Report

Comments and Suggestions for Authors

De Miguel and colleagues provide a manuscript entitled: “Systematic and narrative review of the relationship between mental health and nutrition: The mediating role of interpersonal relationship quality”. The Authors claim that the quality of interpersonal relationships acts as a catalyst and enhancer of the effect of nutrients on mental health. The manuscript is potentially interesting. However, some points need to be addressed.

  • The introduction must be extended beyond the purpose of the review. The Authors must better discuss in general the positive role of specific type of diet for mental health. The Authors wrote: “Several studies have demonstrated a link between diet quality and mental health, contributing to the consolidation of an interdisciplinary field at the intersection of nutrition, psychology, and public health [3-4-5-6-7-8-9-10-11]”. The Authors must extend this part specifying relevant findings. The Authors may also want to consider the following clinical findings suggesting the positive role of specific diet on mental health (DOI: 10.3390/antiox12020272; DOI:10.3389/fnut.2022.943998, and others).

  • The Authors wrote in the first statement of the introduction: “Our study is situated within the field of public health”. I suggest the Authors to substitute the word situated, which is not common in the scientific literature.

  • The Authors must check for typos throughout the manuscript.

  • The Authors must also check for the presence of statements without references throughout the manuscript.

Author Response

Comment 1: The introduction must be extended beyond the purpose of the review. The Authors must better discuss in general the positive role of specific type of diet for mental health. The Authors wrote: “Several studies have demonstrated a link between diet quality and mental health, contributing to the consolidation of an interdisciplinary field at the intersection of nutrition, psychology, and public health [3-4-5-6-7-8-9-10-11]”. The Authors must extend this part specifying relevant findings. The Authors may also want to consider the following clinical findings suggesting the positive role of specific diet on mental health (DOI: 10.3390/antiox12020272; DOI:10.3389/fnut.2022.943998, and others).

Response 1: Thanks for the comment. We have extend the introduction as you can see in lines 44 to 108

Comment 2: The Authors wrote in the first statement of the introduction: “Our study is situated within the field of public health”. I suggest the Authors to substitute the word situated, which is not common in the scientific literature.

Response 2: Thanks for the comment. We've changed the phrase "Our study is situated within the field of public health..." to "Our study is framed within the field of public health..." on line 30.

Reviewer 2 Report

Comments and Suggestions for Authors

The authors present a combined systematic and narrative review of the mediating effects of interpersonal relationship factors on diet and mental health. Overall, the review seems well performed. A few points to consider:

-did you perform a quality or risk of bias assessment per PRISMA guidelines?

-is it possible your systematic review search methodology was too restrictive and therefore was not able to find the papers you were trying to find to answer your question? In other words, should you have used the terms or variants of the terms listed under your qualitative review like history, aesthetics, etc?

-Although I understand your rationale, 23 included studies seems substantial enough for a systematic review alone, perhaps you can re-word your rationale at the beginning of the methods to say the included studies were insufficient in design or methodology to answer your question?

-generally one expects to see a large table describing the included systematic review studies

-generally PRISMA requires a flow diagram per their checklist

Author Response

Comment 1: did you perform a quality or risk of bias assessment per PRISMA guidelines?

Response 1: Yes. A formal assessment of methodological quality and risk of bias was conducted using the Joanna Briggs Institute (JBI) critical appraisal tools, as detailed between lines 217 to 222 and 309 to 332 of the main text and in the complementary material.

Comment 2: is it possible your systematic review search methodology was too restrictive and therefore was not able to find the papers you were trying to find to answer your question? In other words, should you have used the terms or variants of the terms listed under your qualitative review like history, aesthetics, etc?

Response 2: Thanks for the comment. The search strategy was intentionally designed to be broad and inclusive across the three key topics under investigation—mental health, nutrition, and the quality of interpersonal relationships—by incorporating multiple relevant terms for each domain:

- Mental health: using the terms mental health, well-being, and stress;

- Nutrition: using the terms diet, gastronomy, nutrition, food, and nutrients;

- Quality of interpersonal relationships: using the terms interpersonal relationships, socialization, and conviviality.
The structure and rationale behind this selection are detailed between lines 179 and 188 of the manuscrip:

Additionally, it is important to note that all the databases used—PubMed, Web of Science, Scopus, and APA PsycInfo—automatically integrate controlled vocabulary systems, such as MeSH (Medical Subject Headings) in PubMed. MeSH terms are standardized indexing terms assigned to articles to ensure that different expressions referring to the same concept are captured under a unified heading. This guarantees that both specific keywords and their conceptual variants are retrieved, even when different terminology is used in the title or abstract.

Therefore, the combination of specific keywords and the automatic mapping to controlled vocabulary ensures the comprehensiveness of the search strategy.

Regarding other terms such as history or aesthetics, we consider these to represent distinct disciplinary lenses or domains of inquiry rather than essential components of the search string itself. Including them would imply a shift in the scope of the review rather than a broadening of its terminology.

Comment 3: Although I understand your rationale, 23 included studies seems substantial enough for a systematic review alone, perhaps you can re-word your rationale at the beginning of the methods to say the included studies were insufficient in design or methodology to answer your question?

Response 3: Thanks for the comment. As stated in the 3.1.5 section, “The studies included in the systematic review were insufficient in design or methodology to adequately address the research question, which aimed to conceptually understand the articulation of the three core domains: mental health, nutrition, and the quality of interpersonal relationships.” line 463 to 466. Therefore, the study was complemented by a narrative review to address this conceptual gap.

We acknowledge that 23 studies may appear sufficient in number; however, their methodological limitations justified the integration of a complementary narrative approach. This rationale is clarified at the beginning of the Methods section to ensure transparency and alignment with PRISMA guidelines.

Comment 4: generally one expects to see a large table describing the included systematic review studies

Response 4: Thanks for the comment. The table providing detailed descriptions of the studies included in the systematic review is referenced in line 285 to 286 of the main text and is available in the complementary material.

Comment 5: generally PRISMA requires a flow diagram per their checklist

Response 5: Thanks for the comment. The PRISMA flow diagram outlining the study selection process is referenced in line 281 of the main text.

Reviewer 3 Report

Comments and Suggestions for Authors

This is an interesting topic that can help us understand the complex relationship between nutrition and mental health. The following suggestions and questions will be raised for this manuscript to improve its visibility.
1. Keep the title short
2. The first proper noun needs to be displayed in full text.
3. In terms of research methods, the main topics discussed need to be explained in more detail.
For example: interpersonal relationships, mental health, nutrition.
4. Secondly, does the data collected by the study also include "interpersonal relationships, mental health, nutrition"? At least in the current manuscript, it seems that part of the description is missing.
5. In "2.1.1. Databases Consulted and Search Strategy", additional explanations or references are needed to reinforce the importance of the proper nouns selected by the author in different systems.
6. Is there a year limit for literature search? The author needs to provide additional explanations.
7. If the author can draw a flowchart of this study based on the description of the "method" of the manuscript, it will help readers understand the analysis process of the manuscript.
8. Since the author uses three important search knowledge bases to summarize and analyze data, it is puzzling that in the subsequent discussion, several individual works are discussed separately.
Although I can understand that this has a certain relationship with the topics of interpersonal relationships, nutrition, and mental health. But such data collection and presentation feel quite forced.
The author needs to strengthen this point in the introduction and methodology.
Based on the above problem, I think it is more important to draw a flow chart or even an architecture diagram.

In short, the topic of this article is interesting and the method is appropriate, but the author needs to solve the above problems to help readers read more clearly.
I wish you good work and research,

Author Response

Comment 1. Keep the title short

Response 1. Thanks for the comment. The new title is: “Systematic and narrative review of the mediating role of the personal relationships between mental health and nutrition”. Lines 2 and 3

Comment 2. The first proper noun needs to be displayed in full text.

Response 2: Thanks for the comment. Certainly the meaning of the acronym PRISMA needs to be explained and we have done so between lines 130 and PROSPERO on line 154

Comment 3. In terms of research methods, the main topics discussed need to be explained in more detail. For example: interpersonal relationships, mental health, nutrition.

Response 3: Thanks for the comment. In lines 135 to 143 we have tried to provide a framework for the central terms of the article.

Comment 4. does the data collected by the study also include "interpersonal relationships, mental health, nutrition"? At least in the current manuscript, it seems that part of the description is missing.

Response 4: Thanks for the comment. We've clarified the triple interconnection in the introduction (lines 122 to 127) by explaining the scope of the terms (lines 135 to 143). This, along with the new complementary material: search strategies and the response to comment 8, we believe it is clear that articles are required to address the three topics directly.

Comment 5. In "2.1.1. Databases Consulted and Search Strategy", additional explanations or references are needed to reinforce the importance of the proper nouns selected by the author in different systems.

Response 5: Thanks for the comment. In lines 158 to 161 there is a simple reference to the importance of these databases.

These databases were selected due to their methodological rigor, interdisciplinary coverage, and relevance for research on nutrition, mental health, and psychology. PubMed is a leading biomedical database managed by the U.S. National Library of Medicine, widely used in systematic reviews for its comprehensive indexing of clinical and health-related literature. Scopus and Web of Science (WOS) are multidisciplinary citation databases that offer extensive coverage of peer-reviewed journals in the fields of medicine, psychology, social sciences, and public health. APA PsycINFO, managed by the American Psychological Association, is considered the most authoritative source for psychological and mental health research. The selection of these databases aligns with current methodological standards for systematic reviews and ensures comprehensive and high-quality literature coverage (Falagas et al., 2008; McGowan et al., 2016; Moher et al., 2009).

Falagas, M. E., Pitsouni, E. I., Malietzis, G. A., & Pappas, G. (2008). Comparison of PubMed, Scopus, Web of Science, and Google Scholar: strengths and weaknesses. The FASEB Journal, 22(2), 338–342. https://doi.org/10.1096/fj.07-9492LSF
McGowan, J., Sampson, M., Salzwedel, D. M., Cogo, E., Foerster, V., & Lefebvre, C. (2016). PRESS Peer Review of Electronic Search Strategies: 2015 Guideline Statement. Journal of Clinical Epidemiology, 75, 40–46. https://doi.org/10.1016/j.jclinepi.2016.01.021
Moher, D., Liberati, A., Tetzlaff, J., Altman, D. G., & The PRISMA Group. (2009). Preferred Reporting Items for Systematic Reviews and Meta-Analyses: The PRISMA Statement. PLoS Medicine, 6(7), e1000097. https://doi.org/10.1371/journal.pmed.1000097

In response to your comment, we realized that we haven't provided detailed search strategies for each database. In line 176 to 177, we indicated that the complete strategy can be found in the Complementary Material: Search Strategies.

Comment 6. Is there a year limit for literature search? The author needs to provide additional explanations.

Response 6: Thanks for the comment. The last search was conducted on 25th March 2025. See line 172.

Comment 7. If the author can draw a flowchart of this study based on the description of the "method" of the manuscript, it will help readers understand the analysis process of the manuscript.

Response 7:Thanks for the comment. The PRISMA flow diagram outlining the study selection process is referenced in line 281 of the main text.

Comment 8. Since the author uses three important search knowledge bases to summarize and analyze data, it is puzzling that in the subsequent discussion, several individual works are discussed separately.

Although I can understand that this has a certain relationship with the topics of interpersonal relationships, nutrition, and mental health. But such data collection and presentation feel quite forced.

The author needs to strengthen this point in the introduction and methodology.

Based on the above problem, I think it is more important to draw a flow chart or even an architecture diagram.

Response 8:

In the introduction (lines 122 to 127), we have further explained the intention, pointing out that the methodology is what ensures searches where the three central themes intersect in each article. In the updated methodology section, this is fully explained, as we have added supplementary material on search strategies.

Before the exposition of the three axes, we have added (lines 345-351): “All the articles reviewed address the three core themes: mental health, interpersonal relationships, and nutrition. However, for the sake of greater analytical clarity and to deepen the understanding of how these dimensions interact, we have chosen to present the findings according to thematic pairs (nutrition and mental health, relationships and mental health, nutrition and relationships). This structure allows for a more detailed exploration of each connection, gradually contributing to a more integrated and comprehensive picture of the phenomenon under study.”

We have also added a table in line 692 showing how the different themes intersect in the narrative review.

Round 2

Reviewer 1 Report

Comments and Suggestions for Authors

The Authors have addressed all the points I raised. I suggest to check the correct order (numbers) of the references in the manuscript.

Reviewer 3 Report

Comments and Suggestions for Authors

I would like to thank the author for taking the comments seriously and submitting a revised manuscript. I think this revised manuscript has been improved. I suggest the editor proceed to the next step.